# Pattern Recognition of EMG Signals by Machine Learning for the Control of a Manipulator Robot

**DOI:** 10.3390/s22093424

**Published:** 2022-04-30

**Authors:** Francisco Pérez-Reynoso, Neín Farrera-Vazquez, César Capetillo, Nestor Méndez-Lozano, Carlos González-Gutiérrez, Emmanuel López-Neri

**Affiliations:** Centro de Investigación, Innovación y Desarrollo Tecnológico UVM (CIIDETEC-UVM), Universidad del Valle de Mexico, Querétaro 76230, Mexico; francisco.perez@uvmnet.edu (F.P.-R.); nein.farrera@uvmnet.edu (N.F.-V.); cesar.capetillo@uvmnet.edu (C.C.); nestor.mendez@uvmnet.edu (N.M.-L.); calberto.gonzalez@uvmnet.edu (C.G.-G.)

**Keywords:** EMG, pattern recognition, machine learning, robot, cyber-physical systems

## Abstract

Human Machine Interfaces (HMI) principles are for the development of interfaces for assistance or support systems in physiotherapy or rehabilitation processes. One of the main problems is the degree of customization when applying some rehabilitation therapy or when adapting an assistance system to the individual characteristics of the users. To solve this inconvenience, it is proposed to implement a database of surface Electromyography (sEMG) of a channel in healthy individuals for pattern recognition through Neural Networks of contraction in the muscular region of the biceps brachii. Each movement is labeled using the One-Hot Encoding technique, which activates a state machine to control the position of an anthropomorphic manipulator robot and validate the response time of the designed HMI. Preliminary results show that the learning curve decreases when customizing the interface. The developed system uses muscle contraction to direct the position of the end effector of a virtual robot. The classification of Electromyography (EMG) signals is obtained to generate trajectories in real time by designing a test platform in LabVIEW.

## 1. Introduction

A person with a disability is an individual who has one or more physical or mental deficiencies that prevent their full and effective participation in equal conditions when interacting with different social environments. In recent years, the development of HMI for people with motor disabilities has been oriented towards the use of systems based on Electromyography (EMG). In [1], a review of the state of the art in EMG monitoring is presented in terms of applications in rehabilitation and minimally invasive acquisition devices; among the advantages that it highlights are in the fields of physiotherapy and telemedicine. In [2], through three EMG channels, they control the position of a robot with two degrees of freedom; the processing is done as a function of time through the amplitude of the signal when movements are made with the elbow and the shoulder joint. Four channels of surface electromyography acquisition are proposed in [3], where pairs of electrodes are placed according to the position and orientation of the target muscles. Selecting materials with excellent properties for devices on the skin, the fabricated electrodes achieve low skin electrode impedance and record sEMG signals with a high signal-to-noise ratio. In [4], a review on signal acquisition and pattern recognition through Machine Learning is presented. In [5], a myoelectric pattern recognition-driven hand exoskeleton was designed for stroke rehabilitation. It detects and recognizes the intention of movement based on EMG signals, and then the exoskeleton helps the user to perform six types of hand movements in a real way. One of the main challenges in the design of interfaces based on sEMG is the obtention of a signal function or model that allows for the reliable control of a care system. Due to the non-stationary signal behavior, three methods are generally used for sEMG analysis to extract information, which are in the time [6,7], frequency, and time–frequency domain [8]. There are some practical factors, such as the change in arm position, that prevent robust myoelectric control. In [9], an experiment with 14 subjects is carried out to accurately characterize factors that alter the EMG recording. Using regression algorithms, they obtain real-time feedback on changes in the position of the arm and displacement of the electrodes. Pattern recognition has been studied further to develop control algorithms for electric hand prostheses [10,11]. These works have shown excellent accuracy when classifying different types of hand movement (>95% for 10 classes), [12,13,14]. Most of the pattern recognition approaches have the limitation that only one of the functions of the prosthetic hand can be controlled, due to its sequential and binary control. Such control strategies make it impossible to perform natural movements of the hand that consist of the simultaneous activation of different degrees of freedom. Some studies have introduced new pattern recognition schemes that classify combined movements [15,16,17,18,19]. The disadvantage of the new approach is the total number of classes, as it increases drastically when new classes are considered. Recently, regression-based approaches have been on the rise, as they provide control information that allows for multi-degree-of-freedom control. In this work, a regression algorithm using neural networks is proposed to obtain a model through multiclass categorization that allows for the control of a robotic system with three degrees of freedom of the anthropomorphic type. The analysis of a single channel of sEMG that classifies signals with different times of muscle contraction is implemented, with the objective that the robot moves accurately according to predetermined positions in a state machine and demonstrates the correct operation of an HMI by reducing the learning curve. In [20], a study of multichannel electromyography signals is carried out, which is one of the methods used in the recognition of human movement patterns. An exoskeleton robot is controlled and EMG signals are measured during dynamic or isometric muscle contractions. As a result, they developed a pattern recognition model of dynamic and isometric muscle contractions using the Short Time Fourier Transform (STFT).

Section 2 presents the fundamentals of the EMG signal. Section 3 presents the design of the HMI from the acquisition of the EMG signal and its analog and digital processing. The multiclass classification model obtained using neural networks is described. The different classes to be detected, the training algorithm and the operation of a state machine that determines the position of the robot according to the result obtained from the model are shown. The classification to reach the desired position is explained, obtaining the dynamics and inverse kinematics of an anthropomorphic robot with three degrees of freedom. A PD+ control is implemented to apply the necessary torque to each joint of the robot and to validate the operation of the HMI. It designs a graphical user interface in LabVIEW software by interacting a virtual robot and the EMG signal. Section 4 describes the results obtained with the classifier, the experimental tests and the response time for each test.

## 2. HMI Systems Based on EMG

The neuron is the cellular unit of the central nervous system. It has two properties: (1) Sensory, which gives it the ability to respond to physical and chemical agents with the initiation of a nerve impulse; and (2) Conductivity, which gives it the property of transmitting these impulses from one side to another. The dendrites that originate in the cell body are responsible for receiving impulses from other neurons and sending them to the soma of their own neuron. The axon is an extension from the neuronal soma that conducts the impulse to the muscle; it is surrounded by a myelin sheath that allows for better impulse conductivity. The neuron that originates the EMG biopotential is called a motor neuron, which conducts the impulse through the neuromuscular junction to the muscle fiber, as shown in Figure 1 [21].

EMG is an electrical exploration of the peripheral nerves by the stimulation of the muscles to achieve their contraction. The differential potential in the biceps brachii is measured by placing two silver/silver chloride (Ag/AgCl) electrodes and a reference electrode located at the junction of the forearm and hand, as shown in Figure 2. [20]. 

When muscle contraction is performed, there are two types: (1) Isometric contraction, which is a static form of exercise in which a muscle contracts to produce force without an appreciable change in muscle length; and (2) Isotonic contraction, which is without appreciable change in the force of contraction. The distance between the origin of the muscle and its insertion becomes smaller. For EMG acquisition study purposes, in the protocol carried out, isometric contractions are recorded by placing a weight in the user’s hand with a value of 5 pounds. This process is carried out in order to avoid the acquisition of noise due to involuntary movements and to keep the arm static while the biceps brachii contraction is performed for short periods of time. This process is carried out in order to avoid the acquisition of noises due to involuntary movements and to keep the arm static while the contraction of the biceps brachii is performed for short periods of recording time, no longer than 45 s, preventing the user from making an unwanted movement due to fatigue. When performing the acquisition, it was observed that the muscle relaxation periods of 5 s made it possible to accurately obtain the muscle contraction times, thus avoiding the introduction of noise due to muscle fatigue. The goal of this work is to demonstrate that, with a correct training of the neural network, adding dynamic muscle contractions due to involuntary movements as an extra class of recognition allows the system to rule out this muscle noise as a motion control command. The EMG signals have amplitudes from 0.1 mV to 5 mV, with a bandwidth of 0 to 5 KHz [21]. With this information, a first acquisition is made using a BIOPAC^®^ commercial system, which allows for the recording of the differential signal taken from two electrodes and a reference, as indicated in Figure 3a. This system records the waveforms of the EMG signal in order to validate the implemented acquisition protocol. Tests were performed for contraction times of 1, 3 and 5 s. In Figure 3b, the response obtained from the EMG signal to a contraction of 5 s with rest pauses also of 5 s is presented. The inconvenience presented in the acquisition protocol when using this system is that the record is stored in a numerical database and cannot be read directly by any other acquisition card. Real-time implementation of the Fast Fourier Transform (FFT) is necessary to verify the spectrum in frequency and obtain the value of the cutoff frequency for the implementation of filters. The next section presents the instrumentation implemented and the digital processing for the acquisition of the database.

## 3. Materials and Methods

The amplifier used is the IC AD620 due to its characteristic of a common rejection ratio of 100 dB and the gain adjustment with an external resistor. A circuit with a basal corrector and a Common Mode Rejection (CMRR) configuration connected to the junction of the forearm and hand is implemented as a circuit reference. According to the amplitude and frequency characteristics of the EMG signal, the analog processing stage is designed, which includes amplification, isolation and filtering.

***A.*** 
*
**Amplification with basal corrector**
*


An instrumentation amplifier CI AD620 is implemented as a preamplification system to acquire the differential EMG signal with a gain of 500. A basal correction circuit is conditioned to eliminate the level of direct current (DC) caused by involuntary movements of the user or an incorrect connection of the electrodes. The circuit is a IC TL084 operational amplifier in its integrator configuration that is connected in feedback to the Ref and $V_{\mathrm{out}}$ outputs of the instrumentation amplifier, as shown in Figure 4, implementing a high pass filter that eliminates the DC bias voltage and preventing op amps from reaching their maximum power limits.

***B.*** 
**
*Analog Filter*
**


To filter the frequency components that are not within the bandwidth of the EMG signal, a range from 0.5 Hz to 5 KHz, a second order bandpass filter in Butterworth configuration with unity gain is designed, with a ratio of 40 dB per decade using high impedance TL084 operational amplifiers, precision resistors and electrolytic capacitors; see Figure 5.

The output of the analog filter stage is connected to the absolute voltage input of a DAQ6009 acquisition card connected via USB port to a laptop, with a sample rate of 10KHz. An acquisition card with a ground plane is designed to decrease inductive noise, as indicated in Figure 6a. Figure 6b shows the response of the acquisition card in the Tektronix^®^ oscilloscope. Analog noise is observed, which is subsequently eliminated by means of a digital filter.

***C.*** 
**
*Digital Filter*
**


Due to the acquisition system being subject to the interference of electromagnetic noise induced by lamps or some other external device, and in order to digitally tune the response of the filter, the design of a digital low pass filter is implemented. First, the analog/digital conversion is done with the National Instrument DAQ6009 card at an acquisition frequency of 10 KHz at 9600 bauds with 11 bits of resolution. The procedure consists of obtaining samples of the continuous signal at instants of time, defining $v_{i}\left[n\right]=v_{n}\left(nT\right)$, where T is the sampling period. 

The response of the digital first order low pass filter is obtained with the aim of reducing the computational cost when applying the filter in real time. The filter configuration is indicated in Figure 7, indicating its response in terms of the complex frequency s. In Equation (1), the filter response is plotted as a function of the complex discrete frequency z.


(1)
Gz=1−eTRCz−11−eTRCz−1


In Equation (2) the filter equation is indicated as a function of the discrete variable n by means of difference equations when implementing the inverse z-transform of Equation (1).
(2)$$v_{o}\left[n\right]=e^{-2{\pi f}_{c}T}{\;v}_{o}\left[n-1\right]+\left(1-e^{-2{\pi f}_{c}T}\right)\;v_{i}\left[n-1\right]$$

To obtain the value of the cutoff frequency $\left(f_{c}\right)$ and tune the digital filter, the Discrete Fourier Transform (FFT) is implemented. First, the EMG signal is digitized by means of a convolution with a Dirac delta pulse train as a function of time, where $v_{i}\left[n\right]$ is a signal represented in an exponential Fourier series, as in Equation (3). $a_{k}$ represents the amplitude of the signal energy.
(3)$$v_{i}\left[n\right]=\overset{\;}{\underset{k=N}{\sum }}a_{k}e^{j\frac{2\pi }{N}\mathrm{kn}}={\;a}_{0}e^{j\frac{2\pi }{N}0n}+a_{1}e^{j\frac{2\pi }{N}1n}\ldots +{\;a}_{N-1}e^{j\frac{2\pi }{N}\left(N-1\right)n}\;$$

The frequency spectrum analysis is performed by applying the Fourier Transform on the discrete signal $v_{i}\left[n\right]$, obtaining as a result a train of delta functions in frequency $X\left(e^{j\varpi }\right)$, as indicated by Equation (4), whose amplitude is determined by the weighting of coefficients $a_{k}$, through the results of the spectrum in Frequency. The component that provides more energy to the signal is calculated; thus, the frequency of the induced noise is determined, and the cutoff frequency is obtained with precision $\left(f_{c}\right)$ for the design of the digital filter. Figure 8 is the result of the implementation of the digital filter in the acquisition of the EMG signal.
(4)$$X\left(e^{j\omega }\right)=\overset{+\infty }{\underset{k=-\infty }{\sum }}a_{k}2\pi \delta \left(\omega -\frac{2\pi }{N}k\right)$$

***D.*** 
**
*Multiclass Classifier: One Hot Encoding*
**


In this section, the method used is presented so that, in real time, the movements determined through the EMG interface are executed on a manipulator robot. An intelligent system for muscle contraction classification was implemented. Using a Multilayer Neural Network (MNN), a model is obtained that identifies four different classes of muscle contraction. The first class is described as Sharp muscle pulse (SMP), the second class as Smooth muscle pulse 3 s (SMP3), the third class as Smooth Muscle Pulse 5 s (SMP5) and, finally, the fourth class is described as Noise Involuntary Movements (NIM). These signals are classified using the One-Hot Encoding technique that labels the waveform of each signal with an integer. Thus, the digital inputs of a state machine are obtained, which determine the predetermined position of a manipulator robot with three degrees of freedom in the Cartesian plane (x, y, z) inside the robot workspace. In Figure 9, the architecture of the HMI based on EMG is presented.

To perform the identification of patterns in the EMG signal of a single channel, they are divided into action potentials with different time intervals. The SMP (Sharp Muscle Pulse) class has an instantaneous contraction interval of 1 s and muscle relaxation intervals of 5 s. The SMP3 (Smooth Muscle Pulse 3 s) class has a contraction interval of 3 s and muscle relaxation intervals of 5 s. The SMP5 (Smooth Muscle Pulse 5 s) class has a muscle contraction interval of 5 s and muscle relaxation intervals of 5 s. The NIM (Noise Involuntary Movements) class is a class that records the resting state of users as well as involuntary arm movements recorded during acquisition. All these samples are stored in a vector called p. Figure 10 indicates the waveform of each class. The SMP, SMP3 and SMP5 classes indicate a position change control order in the manipulator robot, while the NIM class indicates a total stop state, so the MNN has as inputs the different signals identified in classes stored in the vector $p^{1\times n}$. An integer is assigned to each class through supervised training; this labeling is stored in a vector called $T^{1\times n}$, where n is the total number of samples.

***E.*** 
**
*Multiclass Classifier: Multilayer Neural Network*
**


In this section, the implementation of an intelligent system for the classification of EMG signals is presented. The representation of the multilayer neural network is presented in Figure 11, where $p=\left[p^{T}\right]$ is the vector of the R inputs, $b=\left[b^{T}\right]$ represents the polarization of S neurons, $n=\left[n^{T}\right]$ represents the net inputs of each of the S neurons and $W=\left[W_{\mathrm{SR}}{}^{T}\right]$ is the matrix of synaptic weights.

The first stage consists of data normalization because the EMG signals have different voltage thresholds. The description of this procedure is presented in Equation (5), where p represents the data set of the EMG signal by means of a vector of an acquisition channel. The mean of the data is subtracted, with a standard deviation equal to 1 to minimize the computational cost when the network performs the learning process.
(5)$$p=\frac{p-p^{\mathrm{mean}}}{\sqrt{p^{\mathrm{var}}}}=\frac{p-p^{\mathrm{mean}}}{p^{\mathrm{std}}}$$

Algorithm 1 describes the pseudocode for the implementation of the Neural Network in Python; the training consists of assigning to each sample the value of a constant that is stored in the vector T. This vector is the desired result for each class and has the same dimensions as the input vector p.
***Algorithm 1: Multilayer Perceptrón algorithm implemented for the EMG***1 p ←Input_vector
2 T ←Output_vector

3 */*** output vector T where the labeling value is stored by one-hot-encoding of each the classes***/*4 $T\;\leftarrow \left[\left\{0,0,0,1\right\},\left\{0,0,2,0\right\},\left\{0,3,0,0\right\},\left\{4,0,0,0\right\}\right]$
5 $\mathrm{scaler}\;\leftarrow \mathrm{StandardScaler}\left(\;\right).\mathrm{fit}\left(P\right)$
6 $p\;\leftarrow \mathrm{scaler}.\mathrm{transform}\left(P\right)$
7 */**Divide p into a test (*$P_{\mathrm{test}})$
*and a training set*
$\left(P_{\mathrm{train}}\right)$***/*
8 $\mathrm{one}\_\mathrm{hot}\_\mathrm{labels}=\mathrm{to}\_\mathrm{categorial}\left(T,\mathrm{num}\_\mathrm{classes}\leftarrow 4\right)$9 $P\_\mathrm{train},P\_\mathrm{test},T\_\mathrm{train},T\_\mathrm{test}\leftarrow \mathrm{train}\_\mathrm{test}\_\mathrm{split}\left(P,\mathrm{one}\_\mathrm{hot}\_\mathrm{labels},\;\mathrm{test}\_\mathrm{size}\leftarrow 0.20,\mathrm{random}\_\mathrm{state}\leftarrow 42\right)$
10 */**Random Initialization**/*11 $W\leftarrow 2\times \left(\mathrm{random}-0.5\right)\times \mathrm{scale}$
12 epochs←300013 hiddenNodes←4
14 $\mathrm{model}\leftarrow \mathrm{Sequential}\left(\;\right)$
15 $\mathrm{model}.\mathrm{add}(\mathrm{Dense}\left(\mathrm{hiddenNodes},\mathrm{activation}\leftarrow \mathrm{relu},\mathrm{input}\_\dim\leftarrow 4\right)$16 $a\left[1\right]\leftarrow \max\left(0,n\right)$*//ReLu activation function*
17 $\mathrm{model}.\mathrm{add}\left(\mathrm{Dense}\left(4,\mathrm{activation}\leftarrow \prime \mathrm{softmax}\prime \right)\right)$18 $a\left[2\right]\leftarrow e^{n_{4}}/\;\sum_{1}^{5}e^{n_{4}}$*//Softmax activation function*
19 $\mathrm{model}.\mathrm{summary}\left(\;\right)$
20 loss←categorical_crossentropy21 */**Loss function (categorial cross entropy**/*22 $L\left(y,\;\hat{y}\right)\leftarrow \frac{1}{N}\sum_{j=1}^{M}\sum_{i=1}^{N}\left(y_{\mathrm{ij}}\log\left({\hat{y}}_{\mathrm{ij}}\right)\right)$
22 $\mathrm{optimizer}\;\leftarrow \mathrm{tf}.\mathrm{keras}.\mathrm{optimzers}.\mathrm{Adam}\left(\;\right)$24 $W\;\leftarrow W-\frac{\alpha m}{\sqrt{v+\epsilon }}$
25 $\mathrm{model}.\mathrm{compile}\left(\mathrm{loss}\leftarrow \mathrm{loss},\mathrm{optimizer}\leftarrow \mathrm{optimizer},\mathrm{metrics}\leftarrow \left[\prime \mathrm{accuracy}\prime \right]\right)$26 $\mathrm{history}\leftarrow \mathrm{model}.\mathrm{fit}\left(P\_\mathrm{train},T\_\mathrm{train}\;,\mathrm{epochs}\leftarrow \mathrm{epochs},\mathrm{vebose}\leftarrow 1\;,\mathrm{validation}\_\mathrm{split}\leftarrow 0.1\;\right)$
27 $\mathrm{test},\;\mathrm{test}\;\leftarrow \mathrm{model}.\mathrm{evaluate}\left(P,t,\mathrm{verbose}\leftarrow 1\right)$28 $\mathrm{weights}\left(\mathrm{model}.\mathrm{layers},3\right)$29 $\mathrm{scaling}\left(\mathrm{scaler},3\right)$30 $\mathrm{layers}\left(\mathrm{model}.\mathrm{layers}\right)$

In Figure 12, an association between the precision of the neural network with new data $\left(\mathrm{Trian}\;\mathrm{loss}\right)$ and the value of the loss function $\left(\mathrm{Val}\;\mathrm{loss}\right)$ after 3000 epochs is presented. Both graphs have a tendency to zero as training progresses, indicating a correct functioning of the optimizer. In [22], the authors designed multiclass classification on two channels of electrooculography signals and controlled an omnidirectional mobile robot in the X, Y plane. In this work, it is shown that, according to the muscle contraction time, the multiclass classification allows for the control of robotic systems that work in space (X, Y, Z) and that are adaptive to the individual characteristics of the user, achieving a personalization of the Interface.

The obtained values of the synaptic weights *W* and the polarization vector b of the two neurons, after 3000 epochs:$$W_{1}=\left[4\right]\left[1\right]=\left[\begin{array}{c} -0.321\\ 1.016\\ 1.322\\ 1.564 \end{array}\right]$$
$$W_{2}=\left[4\right]\left[4\right]=\left[\begin{array}{c} \begin{array}{cccc} -0.363 & 0.232 & 0.222 & -0.123\\ 0.543 & -0.127 & 0.142 & -0.234\\ 0.126 & -0.123 & -0.118 & 0.233\\ -0.217 & 0.147 & 0.156 & -0.126 \end{array} \end{array}\right]$$
$$b_{1}=\left[\begin{array}{cccc} -0.321 & 0.087 & 0.123 & 0.224 \end{array}\right]$$
$$b_{2}=\left[\begin{array}{cccc} 0.457 & -0.121 & 0.789 & 0.389 \end{array}\right]$$

Once the model recognizes each of the classes by means of integers, a comparison system is implemented using the premise, “If the Network output is: (integer) [1,2,3,4] then 1 is enabled when the network recognizes the waveform that corresponds to each label, otherwise it is 0”. This process allows for a combination of digital pulses for the activation of a state machine.

***F.*** 
**
*State Machine*
**


The combination of digital signals obtained from the pattern recognition of the neural network by means of class classification allows for the transition change of a state machine. A Mealy-type machine is implemented, which generates an output based on its current state and an input. Three finite sets determined by the inputs, outputs and states are defined.

In Figure 13, the transitions of the digital inputs are indicated and the NIM class is represented as the most significant bit. In the next position the SMP3 class is, then the SMP5 class and finally the SMP class, so that there is an input 4 bits for transition change. Each of the states indicates a predetermined position of the manipulator robot with three degrees of freedom in Cartesian coordinates $\left(p_{x},p_{y},p_{z}\right)$. Subsequently, these coordinates are converted to joint coordinates $\left(q_{1},q_{2},q_{3}\right)$ using the inverse kinematics of the manipulator robot. There is an input IN9 that, when detecting a status at 1 of noise or involuntary movements, completely deactivates the operation of the robot; this is taken as a security measure to not activate the robot when this class of signals occurs.

In Figure 14, the designed machine has eight possible states for muscle movement, with four digital inputs corresponding to the high and low pulses of the Neural Network recognition. Table 1 describes the position in Cartesian coordinates of each of the robot states.

The selected robot is an anthropomorphic robot with three degrees of freedom and rotational joints whose operation is similar to the human arm (Figure 15), where $l_{1},l_{2}$ and $l_{3}$ represent the total length of the links, $l_{c1},l_{c2}$ and $l_{c3}$ represent the length from the initial end to the center of mass of each of the links that make up the robot, $m_{1},m_{2}$ and $m_{3}$ are the values of the center of mass of each link, $x_{0\ldots 3},y_{0\ldots 3},z_{0\ldots 3}$ represent the cartesian axes indicating the orientation of the position and $q_{1},q_{2}$ and $q_{3}$ represent each degree of freedom of each rotational joint of the robot.

To determine the workspace of the anthropomorphic robot, the calculation of the forward kinematics is performed, which determines the position of the end effector in Cartesian coordinates $\left(p_{x},p_{y},p_{z}\right)$ based on joint coordinates $\left(q_{1},q_{2},q_{3}\right)$ indicated in Equation (6). These equations are fundamental for the calculation of the robot dynamics.
(6)$$\begin{array}{ccc} p_{x} & = & \cos\left(q_{1}\right)\left(l_{c3}\;\cos\left(q_{2}+{\;q}_{3}\right)+{\;l}_{c2}\;\cos\left(q_{2}\right)\right)\\ p_{y} & = & \sin\left(q_{1}\right)\;\left(l_{c3}\;\cos\left(q_{2}+{\;q}_{3}\right)+{\;l}_{c2}\;\cos\left(q_{2}\right)\right)\\ p_{z} & = & l_{1}+{\;l}_{c3}\;\sin\left(q_{2}+{\;q}_{3}\right)+{\;l}_{c2}\;\sin\left(q_{2}\right) \end{array}$$

Because the state machine has the coordinates of the end effector position in meters for each of the Cartesian axes x,y y z, the inverse kinematics of the robot defined in Equation (7), these equations determine the value of the position in radians for each of the degrees of freedom $\left(q_{1},q_{2},q_{3}\right)$.
(7)$$q_{1}=\tan^{-1}\left(\frac{\mathrm{py}}{\mathrm{px}}\right)\;q_{2}=2{\;\tan}^{-1}\left(\frac{b+\sqrt{b^{2}+a^{2}-c^{2}}}{a+c}\right)$$
where:$$c={\;p}_{x}{}^{2}{\;\cos}^{2}\left(q_{1}\right)+2{\;p}_{x}{\;p}_{y}\;\mathrm{sen}\left(q_{1}\right)\cos\left(q_{1}\right)+p_{z}{}^{2}-2{\;p}_{z}{\;l}_{1}+l_{1}{}^{2}+l_{2}{}^{2}-{\;l}_{3}{}^{2}\;a=2{\;p}_{x}{\;l}_{2}\cos\left(q_{1}\right)+2{\;p}_{y}\;l_{2}\;\mathrm{sen}\left(q_{1}\right)\;b=2{\;p}_{z}{\;l}_{2}-2{\;l}_{1}{\;l}_{2}\;q_{3}=\tan^{-1}\left(\frac{p_{z}\;\cos\left(q_{2}\right)-{\;l}_{1}\;\cos\left(q_{2}\right)-{\;p}_{x}\;\cos\left(q_{1}\right)\;\sin\left(q_{2}\right)-{\;p}_{y}\;\sin\left(q_{1}\right)\;\sin\left(q_{2}\right)\;}{p_{z}\;\cos\left(q_{2}\right)-{\;l}_{1}\;\cos\left(q2\right)-{\;p}_{x}\;\cos\left(q_{1}\right)\;\sin\left(q_{2}\right)-{\;p}_{y}\;\sin\left(q_{1}\right)\;\sin\left(q_{2}\right)}\right)$$

To implement the PD+ position tracking control algorithm, use the dynamic model defined in Equation (8).

Inertia Matrix $\left(M\left(q\right)\right)$


$$\left[\begin{array}{ccc} \left(\begin{array}{c} I_{1}+\;I_{2}+I_{3}+\frac{l_{2}{}^{2}{\;m}_{3}}{2}+\frac{l_{c2}{}^{2}{\;m}_{2}}{2}+\frac{l_{c3}{}^{2}{\;m}_{3}}{2}\\ +\frac{l_{2}{}^{2}{\;m}_{2}\;\cos\left(2q_{2}\right)}{2}+\frac{l_{2}{}^{2}{\;m}_{3}\;\cos\left(2q_{2}\right)}{2}\\ +\frac{l_{c3}{}^{2}{\;m}_{3}\;\cos\left(2q_{2}\;+\;2q_{3}\right)}{2}\\ +\;l_{2}\;l_{c3}\;m_{3}\;\cos\left(2q_{2}\;+{\;q}_{3}\right)\\ +\;l_{2}\;l_{c3}\;m_{3}\;\cos\left(q_{3}\right)\; \end{array}\right) & \left(I_{2}+I_{3}\right) & I_{3}\\ \left(I_{2}+I_{3}\right) & \left(\begin{array}{c} \;I_{2}+\;I_{3}+l_{c2}{}^{2}\;m_{2}\;\\ +{\;l}_{c3}{}^{2}+2\;l_{2}\;l_{c3}\;m_{3}\;\cos\left(q_{3}\right)\\ +l_{2}{}^{2}\;m_{3} \end{array}\right) & \left(\begin{array}{c} \;I_{3}+\;l_{c3}{}^{2}\;m_{3}\;\\ +l_{2}\;l_{c3}\;m_{3}\;\cos\left(q_{3}\right) \end{array}\right)\\ {\;I}_{3} & \left(\begin{array}{c} \;I_{3}+l_{c3}{}^{2}\;m_{3}\;\\ +l_{2}\;l_{c3}\;m_{3}\;\cos\left(q_{3}\right) \end{array}\right) & \left(I_{3}+{\;l}_{c3}{}^{2}{\;m}_{3}\right) \end{array}\right]$$


Coriolis Matrix $\left(C\left(q,\dot{q}\right)\right)$
$$\left[\begin{array}{ccc} \left(\begin{array}{c} -\dot{q_{2}}l_{2}{}^{2}\;m_{2}\sin\left(2q_{2}\right)\\ -\;\dot{q_{2}}l_{2}{}^{2}m_{3}\;\sin\left(2q_{2}\right)\;\\ -\dot{q_{2}}l_{c3}{}^{2}m_{3}\;\sin\left(2q_{2}+2q_{3}\right) \end{array}\right) & \left(-2\dot{q_{1}}{\;l}_{2}{\;l}_{c3}{\;m}_{3}\sin\left(2q_{2}+{\;q}_{3}\right)\right) & \left(\begin{array}{c} -\;\dot{q_{1}}\;l_{c3}{}^{2}\;m_{3}\;\sin\left(2q_{2}\;+\;2q_{3}\right)\\ \;\;-\;\dot{q_{1}}\;l_{2}\;l_{c3}\;m_{3}\;\sin\left(q_{3}\right)\\ -\;\dot{q_{1}}\;l_{2}\;l_{c3}\;m_{3}\;\sin\left(2q_{2}+q_{3}\right) \end{array}\right)\\ \left(\begin{array}{c} \frac{\dot{q_{1}}{\;l}_{2}{}^{2}m_{3}\;\sin\left(2q_{2}\right)}{2}\;\\ +\frac{-\dot{q_{1}}l_{c2}{}^{2}{\;m}_{2}\;\sin\left(2q_{2}\right)}{2}\;\\ +\frac{\dot{q1}\;\mathrm{lc}3^{2}\;m3\;\sin\left(2q2+2q3\right)}{2}\\ +\;\dot{q_{1}}\;l_{2}l_{c3}\;m_{3}\;\sin\left(2q_{2}+{\;q}_{3}\right) \end{array}\right) & \left(-2\dot{q_{3}}{\;l}_{2}{\;l}_{c3}{\;m}_{3}\;\sin\left(q_{3}\right)\right) & \left(-\dot{q_{3}}{\;l}_{2}{\;l}_{c3}{\;m}_{3}\;\sin\left(q_{3}\right)\right)\\ \left(\begin{array}{c} \frac{\dot{q_{1}}{\;l}_{c3}{}^{2}{\;m}_{3}\;\sin\left(2q_{2}+2q_{3}\right)}{2}\\ +\frac{\dot{q_{1}}{\;l}_{2}{\;l}_{c3}{\;m}_{3}\;\sin\left(q_{3}\right)}{2}\;\\ +\frac{\dot{q_{1}}{\;l}_{2}{\;l}_{c3}{\;m}_{3}\;\sin\left(2q_{2}+q_{3}\right)}{2} \end{array}\right) & \left(\dot{q_{2}}{\;l}_{2}{\;l}_{c3}{\;m}_{3}\;\sin\left(q_{3}\right)\right) & 0 \end{array}\right]$$

Gravity Vector $\left(g\left(q\right)\right)$
$$\left[\begin{array}{c} 0\\ -{\;g\;l}_{c3}{\;m}_{3}\;\cos\left(q_{2}+q_{3}\right)-{\;g\;l}_{2}{\;m}_{3}\;\cos\left(q_{2}\right)-{\;g\;l}_{c2}{\;m}_{2}\;\cos\left(q_{2}\right)\\ -{\;g\;l}_{c3}{\;m}_{3}\;\cos\left(q_{2}+{\;q}_{3}\right) \end{array}\right]$$

Viscous friction vector $\left(B\right)$
$$B\dot{q}=\left[\begin{array}{c} B_{1}\dot{q_{1}}\\ B_{2}\dot{q_{2}}\\ B_{3}\dot{q_{3}} \end{array}\;\right]$$

Torque Vector
$$\tau =\left[\begin{array}{c} \tau_{1}\\ \tau_{2}\\ \tau_{3} \end{array}\right]$$
(8)$$\tau =M\left(q\right)\ddot{q}+C\left(q,\dot{q}\right)\dot{q}+g\left(q\right)+B\dot{q}$$
where $M\left(q\right)$ is a positive definite symmetric matrix of n x n called the inertia matrix, with I1,I2,I3 being the moments of inertia of the rigid links of the mechanical structure of the robot, $C\left(q,\dot{q}\right)$ is an n x 1 vector called the vector of centrifugal and Coriolis forces, $B\dot{q}$ is an n x 1 vector that determines the viscous friction, $g\left(q\right)$ is an n x 1 vector of gravitational forces and τ is the n x 1 vector that determines the torques and forces applied by the actuators at the joints.

***G.*** 
**
*Position Control*
**


As a result of the cartesian coordinates $\left(\mathrm{px},\;\mathrm{py},\;\mathrm{pz}\right)$ obtained from the classifier by means of a Multilayer Neural Network and assigned to a discrete event by means of a state machine, the desired Cartesian coordinates for the robot are obtained, which are transformed to joint coordinates $\left(q_{1},{\;q}_{2},{\;q}_{3}\right)$ from inverse kinematics. These values are the inputs for the PD+ type position control system. [15].

The PD+ control with gravity compensation, defined in Equation (9) by τPD+, is an algorithm that includes proportional control of the position error q˜ and velocity error proportional control q˜˙, where Kp,Kv ∈ ℝnxn are the proportional and derivative gains, respectively, both are positive definite matrices, and the full dynamics of the robot are added. In the structure of this scheme, the trajectory of position, velocity and desired acceleration is involved, qdt,qd˙t,q¨dt∈ ℝn.
(9)τPD+=Kpq˜+Kvq˜˙+Mqqd¨+Cq,q˙qd˙+Bqd˙+gq

The objective of this control is to find a torque value, τ, such that it satisfies the expression indicated in Equation (10).
(10)limt→∞q˜q˜˙=00 ∈ℝ2n
where q˜∈ ℝn is the following error and is defined as q˜= qdt−qt, and q˜˙ ∈ ℝn is the velocity error, given by q˜˙=qd˙t−q˙t. Figure 16 indicates the block diagram of the implemented PD+ control.

Figure 17a shows the behavior of the zero-position error trend in each joint coordinate of the robot whose Cartesian coordinate is assigned by the state machine. The operation of the control when reaching the desired joint position is also presented. Figure 17b shows the virtual simulation of the robot applying the PD+ control for the generation of trajectories through the interaction of the EMG signal.

A graphical user interface is designed as indicated in Figure 18b with visual feedback of the EMG signal, the result of the state machine by means of a green indicator that indicates the position detected of the MNN’s classification, the control curves resulting from the implemented PD+ and a simulation of the virtual robot that indicates the position of the end effector. In Figure 18b, the user connection and the operation of the interface to calculate the response time metrics are indicated.

## 4. Results and Discussions

The EMG signal classification method that allows for the generation of coordinates for the trajectory control of a manipulator robot has been developed. The user’s ability to follow a series of point-to-point coordinates previously determined by colors is measured according to the time of sustained contraction. The yellow dot indicates the starting point of the test, the green dots indicate the path to be followed and the blue dot indicates the end point to which the robot’s end effector must reach. Two trajectories are proposed that increase the difficulty indicating a penalty each time the user enters a contraction command other than the one indicated. The time in which the user generates the trajectory is also recorded. The test ends when the user generates the trajectory without penalties. In Figure 19a, the first proposed trajectory is indicated, in Figure 19b, the time and the number of penalties for each test performed by the user are presented and in Figure 19c, a graph of the response time for trajectory 1 is indicated.

A downward trend is observed in this first trajectory in the response time when completing the test with zero penalties. It is shown that 11 repetitions are enough to successfully complete the proposed trajectory. At the beginning, it indicates an initial time of 118.52 s, and, at the end, it indicates an initial time of 77.54 s, which corresponds to a decrease in the response time by 34.58%. In Figure 19a, the second proposed trajectory is indicated. Figure 19b shows the time and the number of penalties for each test performed by the user. Figure 19c indicates a graph of the response time for trajectory 2.

In the second trajectory, a behavior similar to the first trajectory is observed. With 11 repetitions, it is enough to successfully complete the test. At the beginning, an initial time of 188.64 s is indicated, and, at the end, an initial time of 111.99 s is indicated, which corresponds to a decrease in the response time by 40.64%. When performing the test with different points, the same trend is observed in the decrease in response time. By around 11 repetitions, the user has mastery of the HMI. It should be noted that the model is customized for each user according to individual characteristics and muscle contraction time in addition to adding a recognition class for involuntary movements that blocks the operation of the robot and takes it to a “home” state.

## 5. Conclusions

An HMI that allows for the classification of muscular signals according to the contraction time has been designed. The model implemented through a neural network allows for the personalization and classification in real time for the generation of movement commands of a virtual robot. The HMI can be implemented with inexperienced users who need only 11 repetitions to master the operation of the system, reducing the learning curve. The future work of this project is to implement the classification of multiclass signals in a physical robotic system. In assistive systems or bionic prostheses, although there is the limitation that, being a discrete system, the movement command is determined by a state machine, the improvement consists of implementing neurofuzzy systems that allow for the generation of continuous trajectories in the robot. The development of assistance systems through physiological signals is important for people with disabilities since it allows them to better adapt to their work or personal environment.

## Figures and Tables

**Figure 1 sensors-22-03424-f001:**
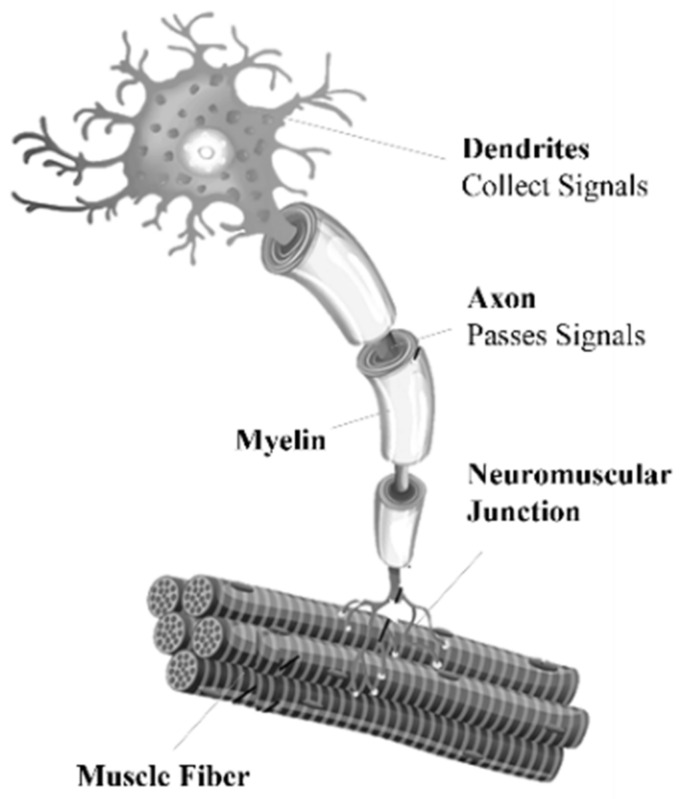
Components of the motor neuron, [21].

**Figure 2 sensors-22-03424-f002:**
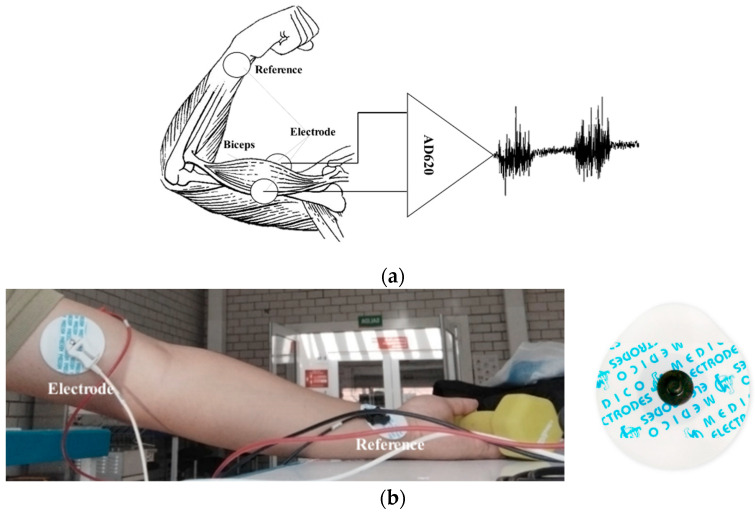
(**a**) Electrode placement diagram and the AD620 instrumentation amplifier, (**b**) Physical representation of the EMG signal acquisition protocol and the Silver/Silver Chloride (Ag/AgCl) electrode implemented.

**Figure 3 sensors-22-03424-f003:**
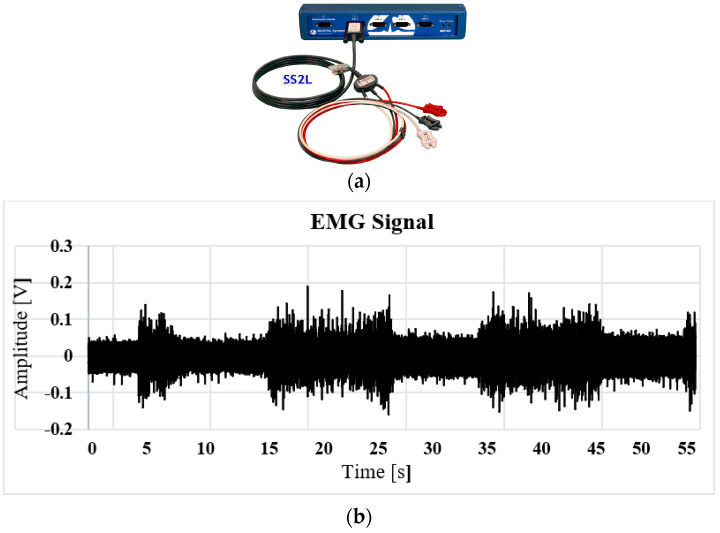
(**a**) Biopac^®^ System, (**b**) Database obtained from the Biopac of the EMG signal with sustained isometric contraction of 5 s.

**Figure 4 sensors-22-03424-f004:**
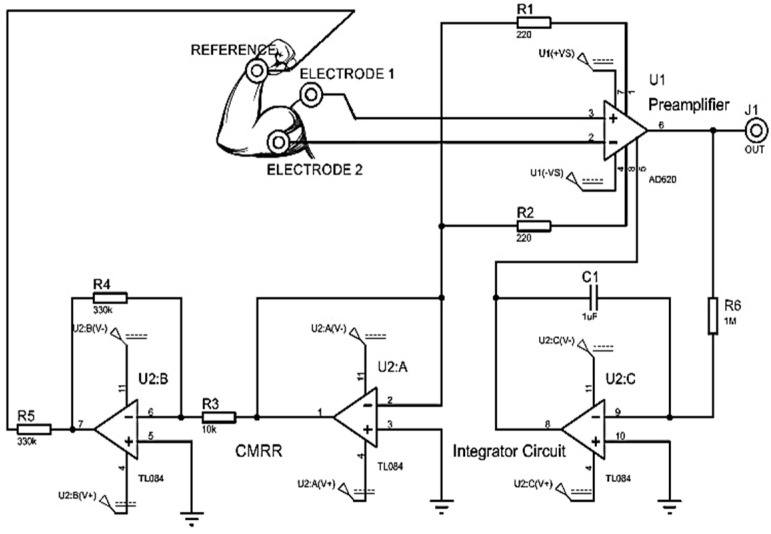
Amplification module and basal corrector.

**Figure 5 sensors-22-03424-f005:**
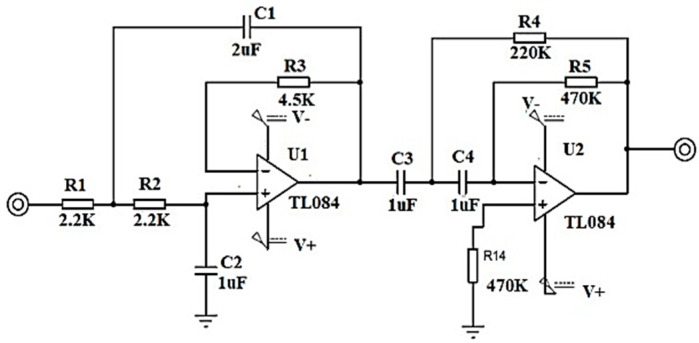
Filter in second order Butterworth configuration at 40 dB/decade.

**Figure 6 sensors-22-03424-f006:**
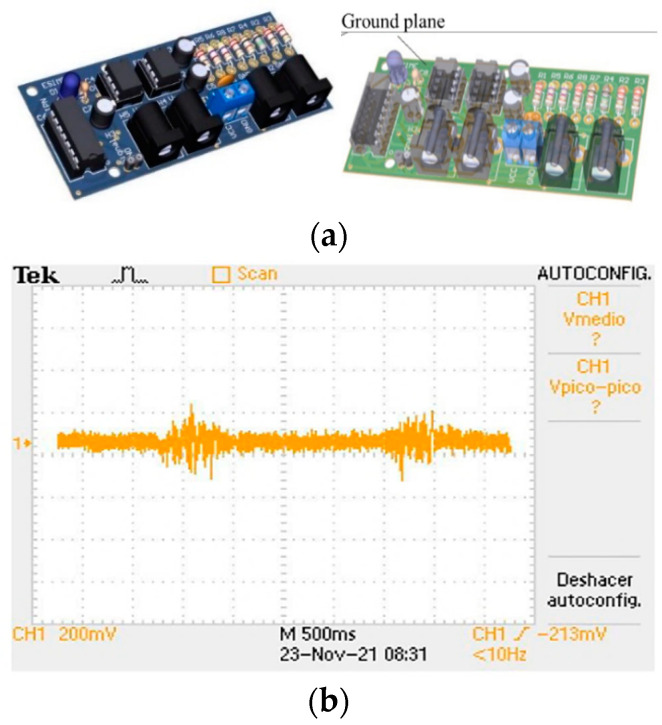
(**a**) EMG signal acquisition cards, (**b**) EMG signal response in the Tektronix oscilloscope.

**Figure 7 sensors-22-03424-f007:**
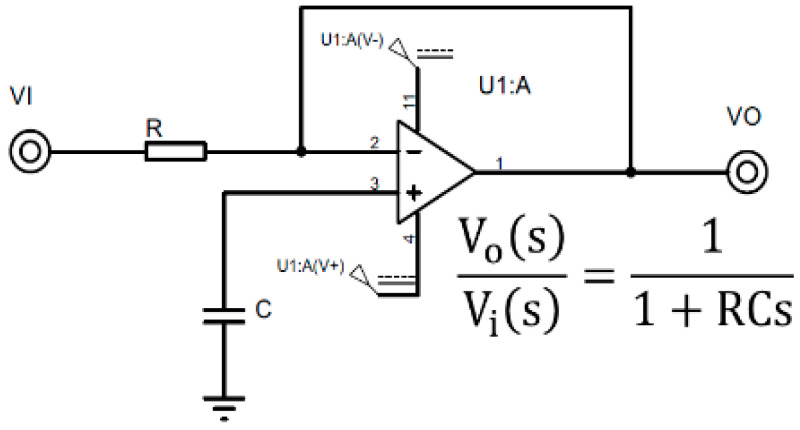
First order low pass filter and its transfer function as a function of the complex variable *s*.

**Figure 8 sensors-22-03424-f008:**
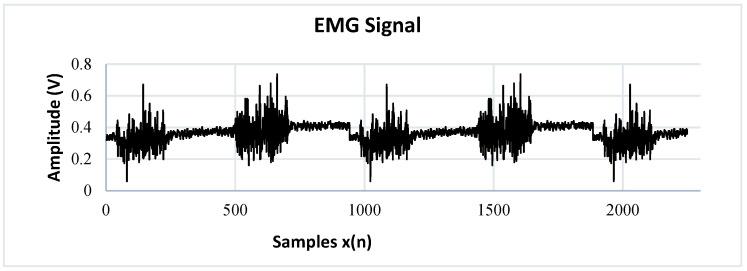
Filtered EMG signal.

**Figure 9 sensors-22-03424-f009:**
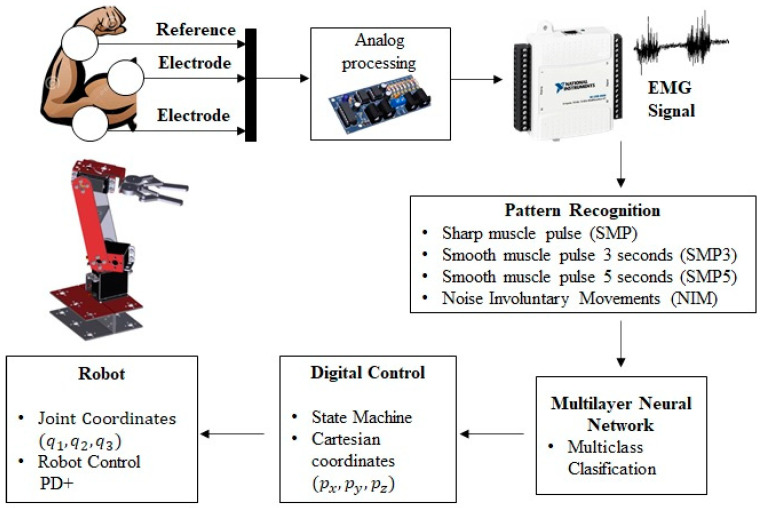
Architecture of the EMG signal classification method for the control of a manipulator robot.

**Figure 10 sensors-22-03424-f010:**
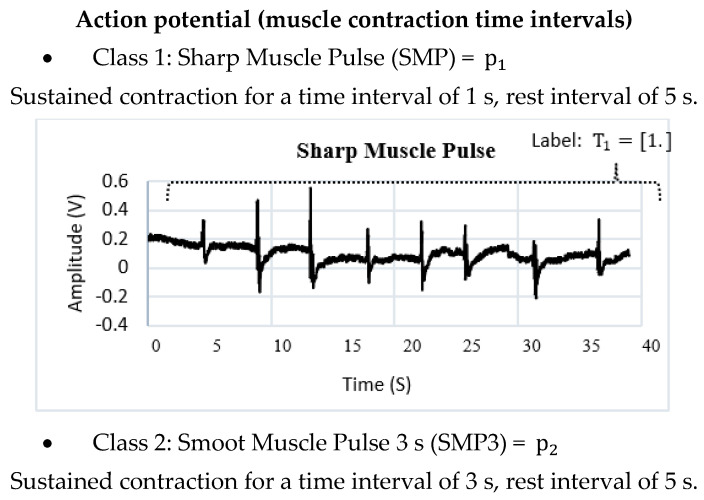
Graphical representation of the data set (input vector p^1×n^). The output vector T^1×n^ stores the labels of each class using integer data.

**Figure 11 sensors-22-03424-f011:**
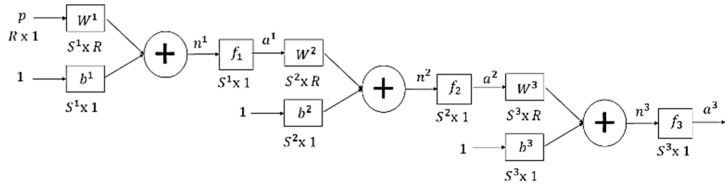
Structure of the Multilayer Neural Network.

**Figure 12 sensors-22-03424-f012:**
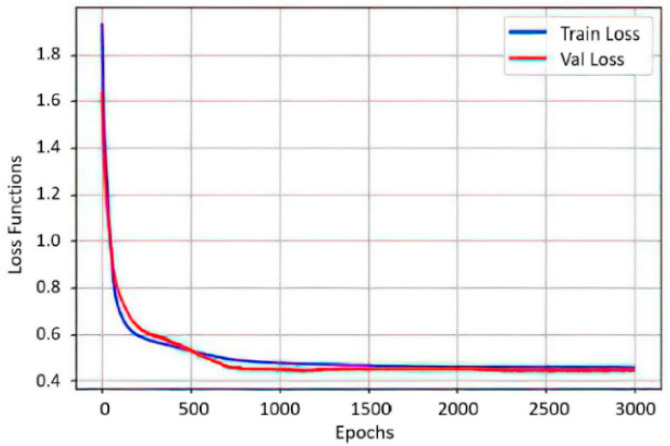
Graph of the accuracy trend of the neural network with new data (Train loss) and the trend of the loss function (Val loss). Neural network accuracy ratio after 3000 epochs.

**Figure 13 sensors-22-03424-f013:**
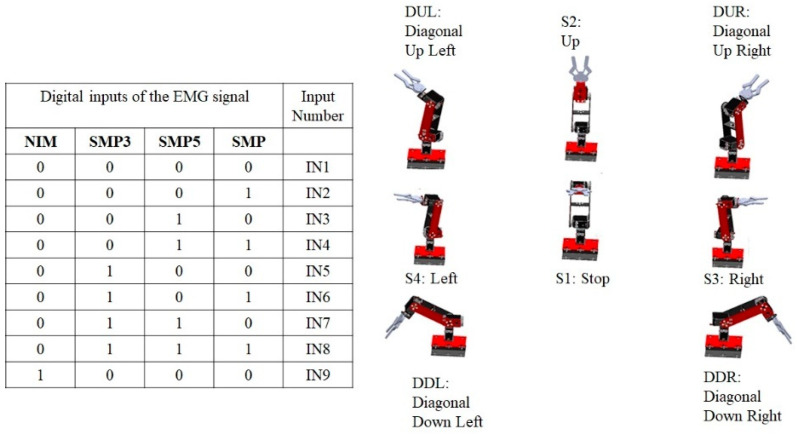
The table presents the inputs of the digital system, and the system outputs are indicated by means of the robot diagram.

**Figure 14 sensors-22-03424-f014:**
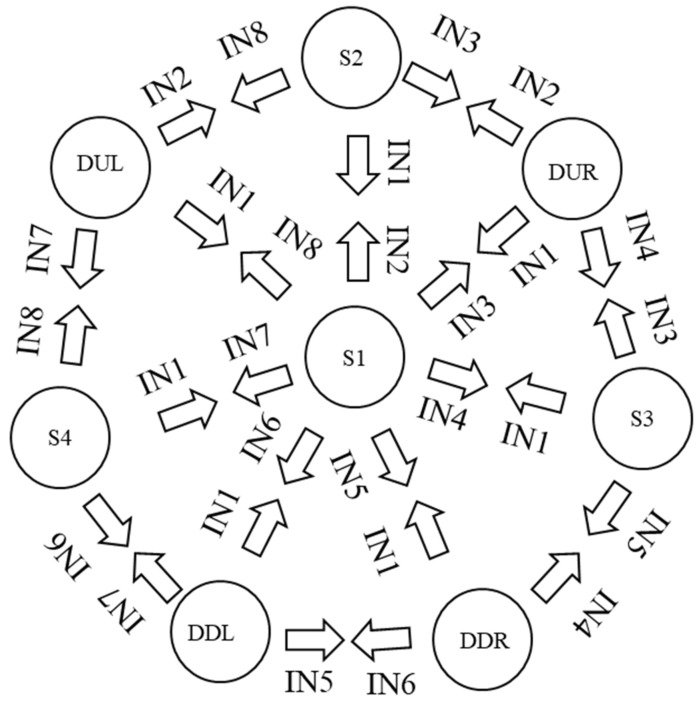
Eight-state Mealy-type machine, transition indicated.

**Figure 15 sensors-22-03424-f015:**
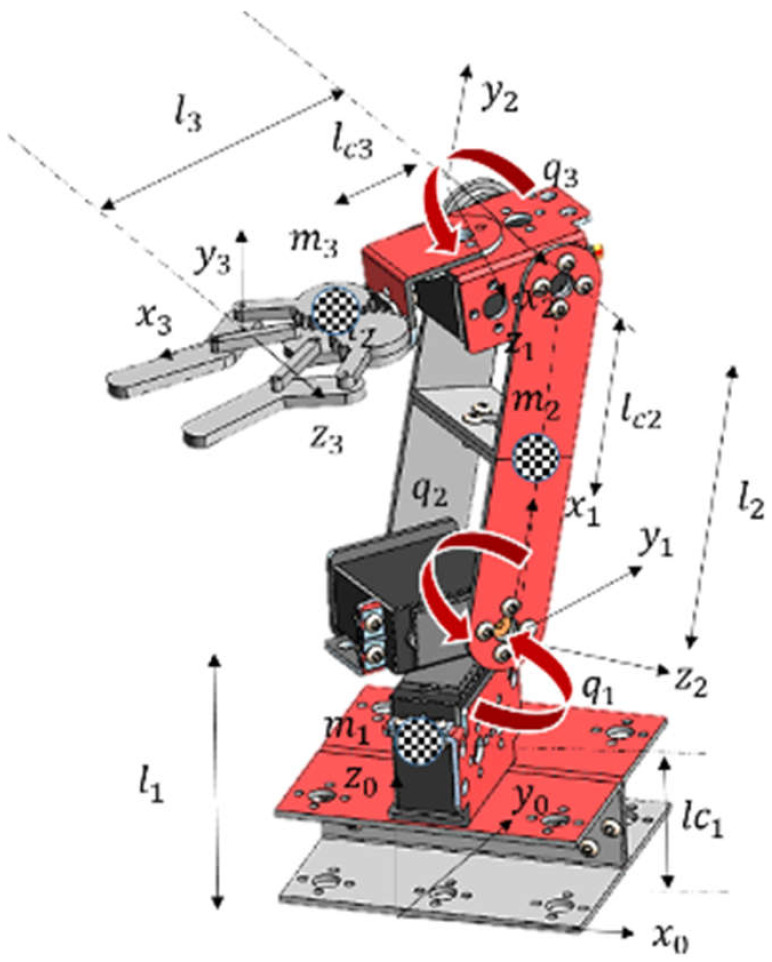
Virtual model of an anthropomorphic robot with three degrees of freedom.

**Figure 16 sensors-22-03424-f016:**
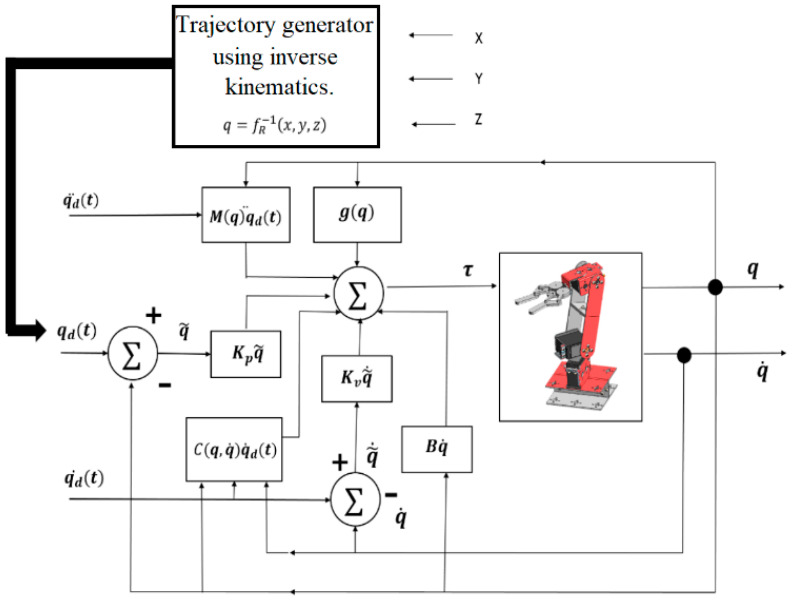
Block diagram of PD + control with gravity compensation.

**Figure 17 sensors-22-03424-f017:**
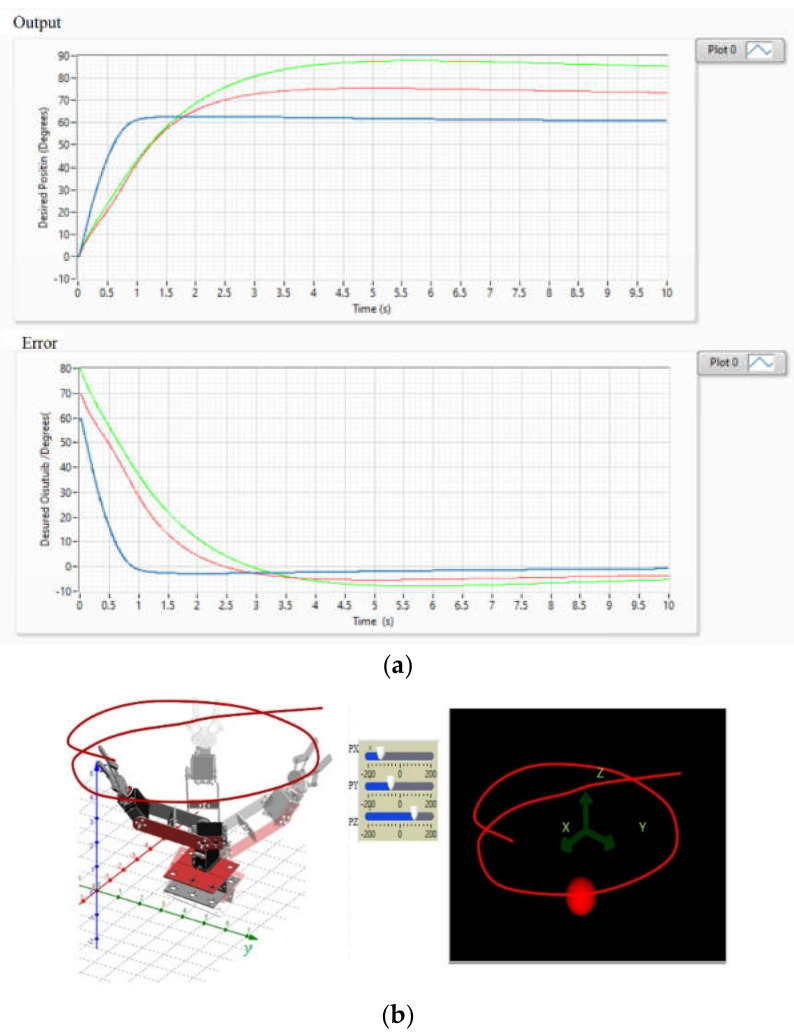
(**a**) Graphs of the position error with a tendency to zero for each of the joint coordinates $\left(q_{1},q_{2},q_{3}\right)$ and control curves indicating the operation of the PD+ to reach the desired positions, (**b**) Result of the PD+ trajectory control of an anthropomorphic virtual robot.

**Figure 18 sensors-22-03424-f018:**
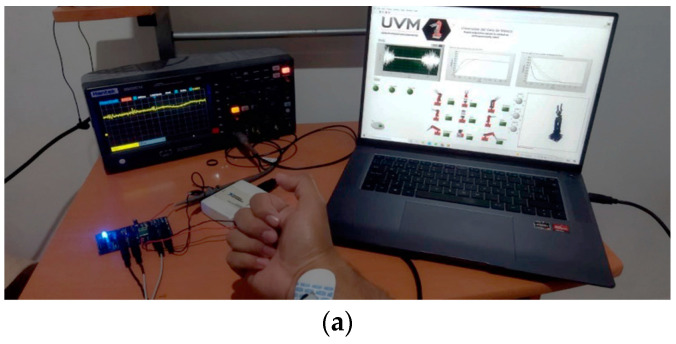
(**a**) Implementation of the real-time acquisition system interacting with the virtual robot simulation, (**b**) Graphical interface designed to record response time metrics.

**Figure 19 sensors-22-03424-f019:**
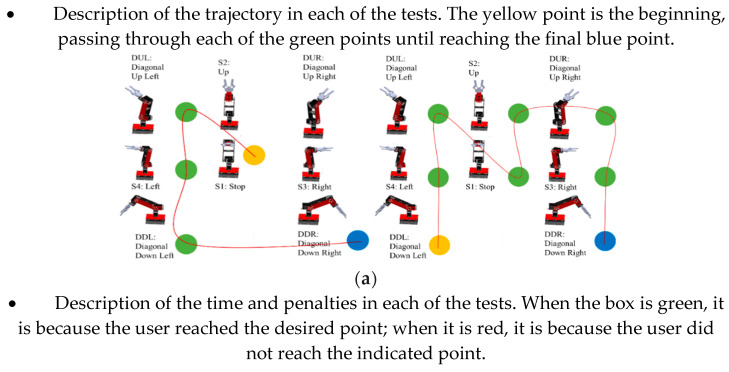
(**a**) Point-to-point trajectories (Trajectory 1 and Trajectory 2), (**b**) The time and the number of penalties (Trajectory 1 and Trajectory 2) and (**c**) Plot of trend response for each trajectory (Trajectory 1 and Trajectory 2).

**Table 1 sensors-22-03424-t001:** Description of each of the desired positions of each state.

State	Input EMG	Desired Value in Meters			Desired Movement
		$p_{x}$	$p_{y}$	$p_{z}$	
**S1**	IN1	0	−0.34	0.38	*Stop*
**S2**	IN2	0	−0.11	0.46	*Up*
**S3**	IN4	0.34	−0.34	0.38	*Right*
**S4**	IN7	−0.34	−0.34	0.38	*Left*
**DUL**	IN8	−0.34	−0.11	0.46	*Diagonal Up Left*
**DUR**	IN3	0.34	−0.11	0.46	*Diagonal Up Right*
**DDL**	IN6	−0.34	−0.34	0.28	*Diagonal Down Left*
**DDR**	IN5	0.34	−0.34	0.28	*Diagonal Down Right*

## Data Availability

All datasets generated for this study are included in the paper.

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
