# Peer review of "Pattern Recognition of EMG Signals by Machine Learning for the Control of a Manipulator Robot"

_sensors, 2022, doi:10.3390/s22093424_

Round 1
Reviewer 1 Report
- The full text mentioned HMI many times, but the full name did not appear. It is recommended to add the full name in the abstract section.
- Line 73 mentioned reference 20 but did not compare the differences between this research and reference 20.
- Line 129 mentioned EMG's signal acquisition system, but did not introduce the related parameters such as its sampling rate.
- Lines 133-134 mentioned “5 seconds of contraction and 5 seconds of rest”, which is confusing, please explain in detail.
- The part about amplifiers and filters in Material and methods Section is a little bit long, please reduce the relative part to highlight multi-class classification part.
- Line 313 mentioned IN9, and the specific definition was presented in Table 1 below. It is recommended to introduce the meaning of IN9 when it is mentioned for the first time.
- The legends of Figures 5 and 8 were not in English
- The horizontal axes of Figures 8 and 10 have no scale.
- Figures 10, 17, and 18 were not clear enough, and the fonts in some of the figures were too small.
- Figures in the Results and discussion Section have no legends and were not clear enough, “11 repetitions” were only mentioned in line 424, but not brought up in the text.
This paper developed a system that applied muscle contraction to control the position of an anthropomorphic manipulator robot. The experiment is novel, but presentation of the experimental procedures needs to be improved. It is recommended to reorganize the related contents to increase readability.
Author Response
We sincerely appreciate your taking the time to review our work. We greatly appreciate each of your comments that help improve the quality of our work. Considering your comments, we have made the following responses:

Reviewer 2 Report
Thank you for the opportunity to review this interesting work. Congratulations to the authors of an interesting work, but I have some comments and questions:
please explain HMI abbreviation
line 95-107 - I propose to delete as too obvious knowledge at the level of general knowledge
line 111 - reference [20] does not correspond to the cited content, I suggest you refer to the seniam.org guidelines,
Please specify what size and type of electrodes were used to record the semg signal
Please describe the study protocol in detail as it is unclear what arm movement was performed by the participants. In natural conditions, we rarely encounter a pure form of isometric or isotonic contraction, most of the muscles work in a mixed cycle. Please explain on what basis was the amount of load for the examined muscle determined?
note Fig. 8 Please use English
I'm sorry, but I don't understand the presented effects of work, maybe, I misunderstood, if the input signal was a static contraction of the biceps muscle, I understand that this type of contraction can cause the robot's work in the sagittal plane, but I don't understand what signal will cause work in the plane frontal and transverse, please explain
Figures 19 and 20 should be presented legibly.
Author Response
We appreciate that you have shared your valuable time and knowledge to review our work. Your respected recommendations have been studied extensively and the responses we have made are shown below.

Round 2
Reviewer 1 Report
The horizontal axe of Figure 3 has no scale, please modify.
Author Response
We sincerely appreciate your taking the time to review our work. We greatly appreciate each of your comments that help improve the quality of our work. Considering your new comments, we have made the following responses:
The horizontal axe of Figure 3 has no scale, please modify.
The horizontal axe of Figure 3 was included